# Association between Aldosterone Synthase (*CYP11B2*) Gene Polymorphism and Hypertension in Pashtun Ethnic Population of Khyber Pakhtunkwha, Pakistan

**DOI:** 10.3390/genes14061184

**Published:** 2023-05-29

**Authors:** Waheed Ali Shah, Asif Jan, Muhammad Asghar Khan, Muhammad Saeed, Naveed Rahman, Muhammad Sajjad Afridi, Fazli Khuda, Rani Akbar

**Affiliations:** 1Department of Pharmacy, University of Peshawar, Peshawar 25000, Pakistan; 2District Headquarter Hospital (DHQH) Charsadda 24430, Pakistan; 3Cardiology Unit Hayatabad Medical Complex, Peshawar 25000, Pakistan; 4Department of Pharmacy, Qurtaba University of Science and Technology, Peshawar 25000, Pakistan; 5Department of Pharmacy, Abdul Wali Khan University, Mardan 23200, Pakistan

**Keywords:** hypertension (HTN), Single Nucleotide Polymorphism (SNPs), whole exome sequencing (WES), genotyping, risk variants, blood pressure (BP), Pashtun, Pakistan

## Abstract

Genome-wide association studies significantly increased the number of hypertension risk variants; however, most of them focused on European societies. There is lack of such studies in developing countries, including Pakistan. The lack of research studies and the high prevalence of hypertension in the Pakistani community prompted us to design this study. Aldosterone synthase (*CYP11B2*) was thoroughly studied in different ethnic groups; however, no such study has been conducted in the Pashtun population of Khyber Pakhtunkhwa, Pakistan. In essential hypertension, the aldosterone synthase gene (*CYP11B2*) plays a significant role. Aldosterone synthesis is affected by both hereditary and environmental factors. Aldosterone synthase (encoded by the *CYP11B2* gene) controls the conversion of deoxycorticosterone to aldosterone and, thus, has genetic influences. Polymorphisms in the *CYP11B2* gene are linked to an increased risk of hypertension. Previous research on the polymorphism of the aldosterone synthase (*CYP11B2*) gene and its relationship to hypertension produced inconclusive results. The present study investigates the relationship between *CYP11B2* gene polymorphism and hypertension in Pakistan’s Pashtun population. We used the nascent exome sequencing method to identify variants associated with hypertension. The research was divided into two phases. In phase one, DNA samples from 200 adult hypertension patients (of age ≥ 30 years) and 200 controls were pooled (n = 200/pool) and subjected to Exome Sequencing. In the second phase, the WES reported SNPs were genotyped using the Mass ARRAY technique to verify and confirm the association between WES-identified SNPs and hypertension. WES identified a total of eight genetic variants in the *CYP11B2* gene. The chi-square test and logistic regression analysis were used to estimate the minor allele frequencies (MAFs) and chosen SNPs relationships with hypertension. The frequency of minor allele T was found to be higher in cases compared to the control (42% vs. 30%: *p* = 0.001) for rs1799998 of *CYP11B2* gene, while no significant results (*p* > 0.05) were observed for the remaining SNPs; rs4536, rs4537, rs4545, rs4543, rs4539, rs4546 and rs6418 showed no positive association with HTN in the studied population (all *p* > 0.05). Our study findings suggest that rs1799998 increases susceptibly to HTN in the Pashtun population of KP, Pakistan.

## 1. Introduction

Hypertension (HTN) is the most frequent modifiable risk factor for cardiovascular disease (CVD), which affects approximately 1 billion individuals worldwide [1,2,3,4]. It is the leading cause of many problems, such as strokes, myocardial infarction, and kidney disease. Despite significant breakthroughs in understanding the pathogenesis of HTN and the development of new treatment options, HTN remains a serious public health concern and has a significant socioeconomic and health cost. Due to its high death rates and absence of early signs, it is also regarded as the world’s most chronic non-communicable illness and, thus, as a “silent killer” [5,6,7,8]. WHO reported that, per year, the hypertension mortality rate is approximately 7.1 million, while low blood pressure management causes 49% and 62% of all ischemic and cerebrovascular heart disease cases, respectively. Appropriate treatment of hypertension can minimize the risk of stroke, renal disease, myocardial infarction and heart failure. The worldwide incidence of HTN in adults is anticipated to rise by 29.2% by 2025 [9]. HTN is more prevalent in quickly developing nations that are experiencing epidemiological transitions, urbanization, economic expansion, and increased life expectancy [10,11]. The burden of cardiovascular disease (CVD) is significant in South Asian nations (Pakistan, Bangladesh, India, Nepal, Sri Lanka, and Bhutan). According to a national health study (2010), HTN affected 33% of individuals aged 45 and 18% of all adults in Pakistan. Moreover, every third hypertensive individual was vulnerable to a wide range of ailments, and only half of the identified patients were adequately taking their medications [12]. According to research performed in May 2019 as part of the International Society of Hypertension (ISH) May Month Measurement (MMM) campaign, the prevalence of HTN in Pakistan is 29.2%. Pakistan also has a high incidence of urbanization, with individuals eating a diet heavy in salts, calories and saturated fats, as well as low in fruits and vegetables [13,14].

Various factors that contribute to high blood pressure were identified, such as environmental, lifestyle and genetic predisposition. Some behavioral factors for hypertension are rapid urbanization, low economic growth and poor lifestyle choices. Like other factors, the genetic makeup of an individual is also a main contributing factor for high blood pressure. Studies on gene analysis identified various chromosomal loci that encourage susceptibility to hypertension. HTN is a highly heritable disease, with additive genetic factors accounting for 50% to 60% of blood pressure (BP) variability. The sum of rare and common genetic variants robustly identified through linkage and genome-wide association studies (GWAS) explains only 1 to 2% of variation in blood pressure (BP) within populations, and the paucity of signals from large-scale GWAS of BP traits indicates that part of the missing heritability is likely to lie in additional undiscovered common or low-frequency variants not currently captured in GWAS SNP arrays. HTN is a complex disease that is impacted by both genetic and environmental factors [15,16]. The renin–angiotensin–aldosterone (RAAS) system regulates BP, as well as fluid and electrolyte balance [17]. As a result, polymorphisms in RAAS candidate genes were extensively analyzed by previous studies based on prior knowledge of biological activities, having the objective of determining the impact of RAAS genetic variability on HTN. The extracellular volume, blood pressure and plasma sodium content are all maintained through RAAS. Several pathologies may arise from an imbalance in this system; cardiovascular diseases, including essential hypertension, myocardial infarction and hypertrophic cardiomyopathy, are linked to molecular variations in genes that encode components of the renin–angiotensin–aldosterone system. One of these is aldosterone synthase, which is a P450 mitochondrial oxidase that is primarily found in the zona glomerulosa of the adrenal medulla and encoded using the *CYP11B2* gene (*CYP11B2*). The aldosterone synthase gene (*CYP11B2*) is located close to the 11-hydroxylase-encoding *CYP11B1* gene. The steroid 11-hydroxylase, 18-hydroxylase and 18-oxidase activities found in aldosterone synthase are necessary for the final step of aldosterone biosynthesis. Although variations in the activity of the related enzyme 11-hydroxylase *(CYP11B1*) can also affect the biosynthesis of steroid metabolites with mineralocorticoid-like effects, cortisol is primarily produced by this enzyme. However, the molecular mechanism underlying this association is unknown. Polymorphic variation in *CYP11B2* is linked to an increased chance of hypertension. The *CYP11B2* gene was previously found to have a number of common polymorphisms. These polymorphisms are linked to hyperaldosteronism, plasma glucose and glucose intolerance, type II diabetes mellitus (DM), left ventricular size and mass, arterial stiffness, myocardial infarction and hypertension. While the roles of RAAS genes, such as angiotensin-converting enzyme (*ACE*), angiotensinogen (*AGT*), angiotensin II receptor type 1 (*AGTR1*) and aldosterone synthase (*CYP11B2*) in hypertension were previously extensively studied in various ethnicities, few studies conducted regarding the Pashtun population exist [18,19,20,21,22,23,24]. 

Essential HTN is a complex condition, and the lack of a single identified etiology further complicates treatment. Obesity, salt, alcohol intake, physical exercise and chronic stress are all controllable variables that influence BP values. Nonetheless, family studies proved that hereditary factors account for 30–50% of a person’s risk, and a family history of HTN increases the likelihood of experiencing HTN four-fold [25,26,27,28]. Many investigations were undertaken over the last several decades to determine the genetic etiology and related molecular mechanism of HTN. BP is significantly influenced by the genes involved in Mendelian types, as determined via linkage analysis and next generation sequencing [29]. A similar approach may be used to determine the pathophysiology mechanism of a specific disease, leading to a specific therapy [30]. One such strategy is GWAS, which is based on families, twins, adopted children or populations. This research can identify single nucleotide polymorphisms (SNPs) that alter the chance of experiencing HTN [31]. 

The *CYP11B2* (aldosterone synthase) genes are definitely linked to common kinds of HTN, according to human and animal studies. The locus was linked to a number of rare but significant monogenic syndromes associated with cardiovascular dysfunction. Previous investigations on the role of *CYP11B2* in essential HTN were limited and underpowered, with the majority focusing on a polymorphism in *CYP11B2*’s 5′ regulatory region (44C/T; rs1799998). This variant is closely related to a complex *CYP11B2* polymorphism that replaces a chunk of the so-called wild-type (Wt) sequence with the corresponding region of *CYP11B1*. Although there is substantial diversity among research, rs1799998 polymorphism is definitively linked to HTN [2,32,33,34,35].

The study of epigenetic changes recently gained popularity, with DNA methylation, histone modification and non-coding RNAs all found to play important roles in causing a variety of pathophysiological processes, including BP control [36]. A variety of disorders are linked to high BP induced through a single uncommon gene mutation. The identification of genes responsible for monogenic types of HTN established the kidney and adrenal glands as key organs in the control of BP [37,38]. The majority of these diseases are caused by mutations that produce abnormalities in the mineralocorticoid, glucocorticoid or sympathetic systems. The most prevalent condition that changes the mineralocorticoid pathway is familial hyperaldosteronism type 1 (FH1) or glucocorticoid–remediable hyperaldosteronism (GRA). This condition is caused by an uneven crossing over of CYP11B1 (encoding 11-hydroxylase) and *CYP11B2* (encoding aldosterone synthase [39].

Pakistan’s population is divided into five ethnic groups: Punjabis, Pashtuns, Sindhis, Baluchis, and Muhajirs. The Pashtuns constitute the bulk of Khyber Pakhtunkhwa’s population (KP). They have different cultural traditions, social beliefs, lifestyles and behaviors that make them a suitable group for this type of study. The genetic mutation spectrum of HTN in the Pakistani population is thought to differ from that of other populations [40]. This work used high-throughput sequencing to identify harmful mutations in the *CYP11B2* gene associated with critical HTN in the Pashtun ethnic group of Khyber Pakhtunkhwa, Pakistan. The study will help us to better understand the pathophysiology of HTN in the study population, as well as the development of modifying techniques to overcome/control the burden of this lethal and costly disease. Through integrating genetic research, phenotypic characterization, longitudinal data and multi-omics approaches, the study can significantly contribute to our understanding of hypertension pathogenesis. The findings can inform the development of targeted interventions, risk stratification tools and personalized approaches to control and overcome the burden of this fatal and costly disease.

## 2. Material and Methods

### 2.1. Subjects Characteristics

A total of n = 400 participants’ age and gender matched (HTN n = 200 and non HTN n = 200) with Pashtun ethnicity and originated from various districts in KP, such as the Peshawar, Mardan, Charsadda, Nowshera, Swabi, Kohat, Bannu, Dir, and Swat districts, were included in this study. Patients were registered at Lady Reading Hospital (LRH) Peshawar; Hayatabad Medical Complex (HMC), Peshawar; or Khyber teaching Hospital (KTH), Peshawar, while control samples were collected from free medical and screening camps located at Rahman Medical Institute (RMI), Hayatabad, Peshawar, and Mardan Medical Complex (MMC), Mardan. The mean (±SE) age of case subjects was 59.56 ± 10.65 (120 males and 80 females), while the control subjects’ mean age was 57.75 ± 12.25 (110 males and 90 females). The study lasted from July 2018 to July 2019. HTN was defined as having a mean BP of 140/90 mmHg or being on antihypertensive medication. Inclusion criteria for cases were as follows: (i) being confirmed as a HTN patient, as per WHO criteria; (ii) being Pakistani and of Pashtun ethnicity; and (iii) being more than 30 years old. Exclusion criteria were as follows: (i) having a mental disorder; (ii) being below 30 years old; and (iii) presenting with a chronic infection, such as HCV, HBS or a malignancy. Control subjects were healthy individuals with a normal blood pressure of 120/80 mmHg. Consent forms and thorough demographic data were created based on carefully designed Proforma. 

### 2.2. Blood Sample Collection

Three milliliter whole blood was taken through applying a aseptic procedure to the median cubital vein of the subjects in the EDTA tube (properly labeled) and stored at −10 °C.

### 2.3. DNA Extraction

DNA was extracted from 200 μL whole blood samples of hypertensive patients using the Wiz Prep DNA extraction kit, by wizbiosolutions (Wiz Prep no. W5400). After extraction, DNA quantification was performed using the Qubit^TM^ ds DNA HS assay kit (Catalog No. Q32851), and the concentration was adjusted to 10 ng/µL [41]

### 2.4. Whole Exome Sequencing

Whole exome sequencing (WES) was performed at the Centre of Genomics, Rehman Medical institute (RMI), Hayatabad, Peshawar. In order to minimize costs and time and simplify sequencing process, DNA pools were constructed from 200 HTN patients and 200 control subjects according to the previously described protocols [42,43]. Each pool contained an equimolar quantity of DNA (100 ng) from each individual. DNA pool amplification and sequencing was carried out via the HiSeq 2500 platform (Illumina, San Deigo, CA, USA) using paired-end libraries (2 × 101) bp [44].

### 2.5. WES Analysis

A custom-built bioinformatics pipeline was used to separate raw sequencing data to create final variants calls. The FASTQ (raw data) file produced using the illumina Hiseq was filtered to remove low-quality reads (*Q* > 30) using CASAVA and the trimometric tool [45,46]. The filtered reads were then aligned to the reference genome (hg19/GRCh37) using BWA-mem (v 0.7.130) [47,48]. Base recalibration was performed using GATAK (v 3.2.2). GATAK unified Genotyper was stored as a VCF FILE. The annotation of the variants was carried out via ANNOVER [49]. The final annotated variants file was run in the Excel program to enable further analysis of the data.

### 2.6. Genotyping of CYP11B2

A total of n = 8 SNPs in CYP11B2 gene were identified using WES. In order to validate whole exome sequencing results and affirm the association between the newly identified CYP11B2 risk variants with HTN, SNPs were genotyped. Genotyping of the selected candidate SNPs was carried at the Centre of Genomics, Rehman Medical Institute (RMI), Hayatabad, Peshawar, using the Sequenom Mass ARRAY^®^ system (Agena Bioscience, San Diego, CA, USA) and following the manufacturer’s guidelines.

### 2.7. Statistical Analysis 

The SPSS application was used to examine statistical data. Age, gender, weight, smoking, lifestyle, exercise and *CYP11B2* gene variations were among the major characteristics investigated. Categorical data from cases and controls were presented as percentages and frequencies, with the Chi-square test used to evaluate them, while continuous variables were presented as mean standard deviations. Using a binary logistic regression model, the odds ratios (OR) of HTN cases for each variation were estimated using 95% confidence intervals (CI).

## 3. Results

### 3.1. Subject Characteristics

Co-morbidities and sociodemographic characteristics of study subjects are shown in Table 1 and Table 2. The prevalence of co-morbidities such as diabetes, hypercholesterolemia, retinopathy and ischemic heart disease were higher in cases than in control subjects, as shown in Table 1. Higher BP > 120/80 mmHg) was observed in patients with HTN, while normal blood pressure was noted in control individuals. Overall, 73% of the study subjects were male and 27% were female. The highest prevalence of HTN patients was observed in Peshawar district (17.5%), followed by Mardan district (15.5), Charsadda district (13%), Dir district (11%), Swabi district (9.5%), Kohat district (9.0%), Bannu district (9.0%), Swat district (7.5%) and Karak district (5%). The patients were from different occupations, such as farming (19%), labor (18%), business (12.5%) and government service (11.5%); a further 15% of patients were retired, while 24% of female patients were housewives. Family history of HTN was recorded in 80.5% of patients, while 17.5% of patients had no family record of HTN. Moreover, 85% of the patients were non-exercising (sedentary life), while 20% had active lives (enjoyed exercising). Most (60%) of the patients used Naswar (a local smokeless tobacco product), while 30% were cigarette smokers. A few patients were conscious about their diet, while a majority made no positive dietary changes.

### 3.2. WES Results

Using WES, we identified a total of 31,743 SNPs, including 3433 homozygous SNPs, 28,309 heterozygous SNPs, insertion, 1200 deletions, exonic, or missense variants; and 58 possibly pathogenic variants, as mentioned in Figure 1. A total of eight SNPs were identified via WES in *CYP11B2*, as described in Table 3.

### 3.3. Minor Allele Frequency Analysis (MAF)

The MAF of rs4536, rs4537, rs4545, rs4543, rs1799998, rs4539, rs4545 and rs6418 for the HTN cases were compared with the control subjects using the chi-square test. A significant difference was observed for rs1799998 (*p* = 0.001) between cases and controls, while the rest SNPs showed no significant difference (*p* > 0.05) in the MAF comparison between the cases and controls, as mentioned in Table 4.

### 3.4. Association between SNP and HTN

All eight WES-identified SNPS were checked for their association with HTN using logistic regression analysis. Among the eight identified SNPS for *CYP11B2*, only the rs1799998 showed positive association with HTN [OR (95% CI); 2.257 (1.76–2.85): *p* = 0.045] in our control study. When the variants were checked in relation to age, gender, smoking and family history of hypertension, the effect of the rs1799998 further increased [OR (95% CI); 2.275 (1.76–2.85): *p* = 0.001). This finding is described in more detail in Table 5.

## 4. Discussion

Aldosterone gene *CYP11B2* polymorphisms are associated with many diseases, such as Type 2 Diabetes Mellitus (T2DM) [50,51,52], myocardial infarction (MI) [53], left ventricular mass [54,55], hyperaldosteronism [56], arterial stiffness [57] and hypertension [58,59,60,61].

A number of studies analyzed associations between T (-344) C of *CYP11B2* and HTN with varying results [62]. A positive association between *CYP11B2* and 344C/T (rs179998) was observed in a study conducted in Han Chinese subjects. This study was based on 17,042 individuals and reported positive associations between -344C/T polymorphism and HTN, as reported in Japanese and European populations [63]. Another interesting study conducted in South Indian Tamils also confirmed noticeable associations between the *CYP11B2*-344C/T (rs179998) and essential HTN [64]. Niu et.al, in a study of Japanese populations, found strong positive associations between *CYP11B2* gene polymorphism and the development of hypertension [65]. Y.R. Kim and colleagues investigated the relationship between variants -344C/T of *CYP11B2* gene and hypertension in Korean patients [66]. A study conducted on the Kazakh community in northwestern China showed polymorphisms of *CY11B2 T-344C* (rs179998) gene and hypertension [67]. Similar results for the *CY11B2 T-344C* (rs179998) gene were also found in populations in European countries, including Belgium, Italy and the United Kingdom [68]. However, many other studies produced contradictory results and discovered either positive association or no link between the *CY11B2 T-344C* (rs179998) gene polymorphism and HTN [69,70,71].

The current study investigated the relationship between *CYP11B2* polymorphism and risk of HTN in the Pashtun ethnic population in Khyber Pakhtunkhwa, Pakistan. The gene was selected for validation due to its prominent association with HTN in other ethnicities [72]. 

We performed WES to identify risk variants associated with HTN in the Pashtun population in Khyber Pakhtunkhwa (KP). It was a multicentre study, and patients were recruited from ten districts in KP. Whole exome sequencing was carried out at Rehman Medical Centre in Peshawar via research collaboration. WES identified total of eight SNPs in the study population. The notable heterozygous variant of *CYP11B2* was rs1799998 located on chromosome 8 at position 142918184. The SIFT and PolyPhen score predicted the variants rs1799998 as a deleterious and probably damaging variant. The minor risk allele T of this variant was found to be higher in these cases compared to the control group (42% vs. 30%). A significant result was found when this SNP was checked for its association with HTN [OR (95% CI) = 2.257 1.76–2.85: *p* = 0.001]. The remaining heterozygous SNPs rs4536 (c. 873G>A), rs4537 (c.842A>G, p.N281S), rs4545 (c.1303G>A, p.G435S), rs4543 (c.891G>A), rs4539 (c.518A>G, p.k173R), rs4546 (c.504C>T) and rs6418 (c.596T>C) showed no association with HTN (all *p* > 0.05) in the study population.

The precise underlying mechanism of *CYP11B2* polymorphism and the incidence of HTN is unknown; thus, large-scale investigations are needed to investigate the mechanism of the rs1799998 variation in HTN.

The socio-demographics of cases and controls shows a higher incidence of DM and hyper-cholestrolemia in these cases compared to the control group. Moreover, the results showed a greater prevalence of family history of HTN in these cases than in the control group (66.5% vs. 25%). Almost half of the affected individuals had a smoking habit. Moreover majority of the cases and controls were inactive and lacked positive exercise habits.

## 5. Conclusions

The current study shows a positive correlation between the -*CYP11B2-* variant rs1799998 and HTN in the Pashtun population in KP; this correlation may be further studied to identify a relevant susceptibility biomarker. Similar studies should be designed on a wide scale to screen people for HTN. Moreover, a public awareness campaign needs to be launched to reduce the prevalence of this “silent killer” disease.

## 6. Limitations

The study’s main limitation is its small sample size. Furthermore, we did not measure the corresponding protein level in order to determine the expression of proteins. Moreover, the current study was conducted on the Pashtun ethnicity; thus, it cannot be assumed to apply to the whole of Pakistan or to other ethnicities.

## Figures and Tables

**Figure 1 genes-14-01184-f001:**
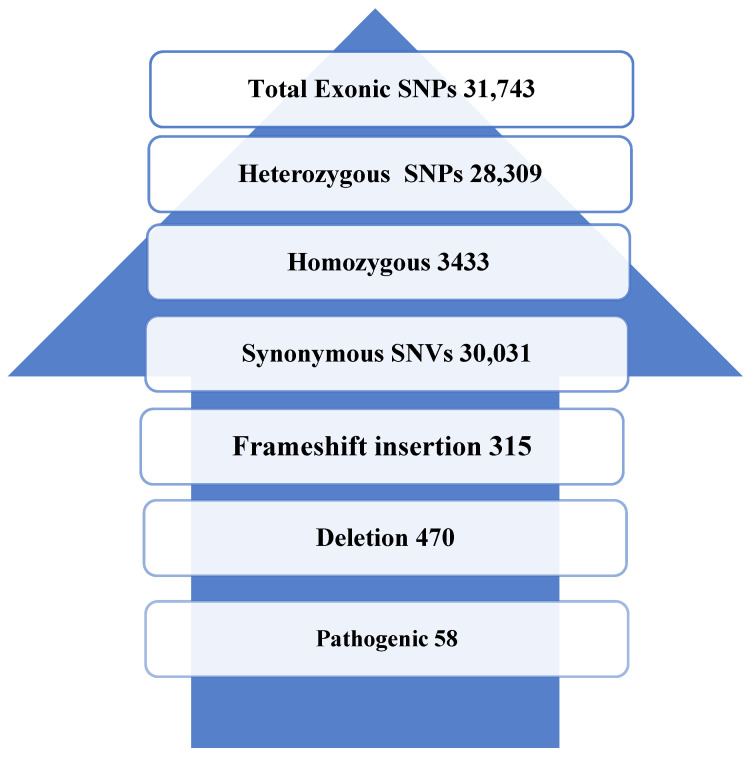
WES identified variants in Pashtun ethnic population (200 HTN cases).

**Table 1 genes-14-01184-t001:** Co-morbidities’ prevalence in study subjects.

Disease	Frequency	
	Cases	Control
Hypercholesterolemia	7.50%	2.83%
DM	28.5%	15.0%
Retinopathy	53.0%	0.00%
IHD	8.00%	0.00%
HCV	0.00%	0.00%
HBV	0.00%	0.00%

DM: Diabetes Mellitus; IHD: Ischemic Heart Disease; HCV: Hepatitis C; HBV: Hepatitis B.

**Table 2 genes-14-01184-t002:** Sociodemographic characteristics of hypertensive patients and controls.

Variables	Cases n (f)	Control n (f)	*p* Value
**Gender**			
Male	130 (65%)	173 (86.5%)	0.051
Female	70 (35%)	27 (13.5%)	
**Address**			
Peshawar	35 (17.5%)	16 (16%)	0.318
Charsadda	26 (13%)	13 (13%)	
Mardan	31 (15.5%)	13 (13%)	
Kohat	18 (9.0%)	11 (11%)	
Swabi	19 (9.5%)	4 (4%)	
Nowshera	17 (8.5%)	5 (5%)	
Bannu	18 (9.0%)	10 (%)	
karak	5 (2.5%)	2 (25%)	
Dir	22 (11%)	10 (2%)	
Swat	15 (7.5%)	10 (10%)	
**Occupation**			
Business	30 (15.0%)	6 (6.0%)	0.058
Govt. servant	37 (18.5%)	27 (27.0%)	
Retired	35 (17.5.0%)	30 (30.0%)	
Farming	25 (12.5%)	10 (10.0%)	
House wife	40 (20.0%)	15 (15.0%)	
Labor	33 (16.5%)	12 (12.0%)	
**Family history of HTN**			
Yes	133 (66.5%)	25 (25%)	0.003
No	67 (33.5%)	75 (63%)	
**Exercise**			
Yes	71 (35.5%)	43 (34%)	0.128
No	129 (64.5%)	57 (57%)	
**Smoking**			
Yes	96 (48%)	20 (20%)	0.093
No	104 (52.0%)	80 (80%)	
**Naswar**			
Yes	130 (65.0)	53 (53%)	0.081
No	70 (35.0%	47 (47%)	
**Diet and drug compliance**			
Yes	127 (63.5.5%)	42 (42%)	0.212
No	73 (36.5)	58 (58%)	
**Socioeconomic factors**			
Good	52 (26.0%)	34 (34%)	0.314
Average	102 (51.0%)	53 (53%)	
Below	46 (23%)	13 (13%)	

Abbrevation: n: patients; f: frequency.

**Table 3 genes-14-01184-t003:** WES identified SNPs of *CYP11B2* gene in study population.

SNP ID	Gene	Variant	Chr Position	Sift Prediction	Polyphen Prediction	Minor Frequency (%)		Read Depth	
						Cases = 200	Control = 200	Cases	Control
rs4536	*CYP11B2*	C>T	8:143995761	Tol	Benign	0.10	0.09	130	120
rs4537	*CYP11B2*	T>C	8:143995792	Tol	Benign	0.23	0.22	100	110
rs4545	*CYP11B2*	C˃T	8:143994041	Tol	Benign	0.08	0.06	76	70
rs4543	*CYP11B2*	C˃T	8:143995743	Tol	Benign	0.04	0.03	150	140
rs1799998	*CYP11B2*	T˃C	8:142918184	Del	Prob Dam	0.42	0.30	250	210
rs4539	*CYP11B2*	T>C	8:143996539	Tol	benign	0.34	0.35	50	40
rs4546	*CYP11B2*	G>A	8:143996553	Tol	Benign	0.25	0.23	40	38
rs6418	*CYP11B2*	A>G	8:143996363	Tol	Benign	0.06	0.05	30	25

Reference genome assembly: hg19/GRCh37; Abbreviations: SNP: single nucleotide polymorphism: Chr; chromosome: prob. dam.; probably damaging: Del; deleterious.

**Table 4 genes-14-01184-t004:** MAF comparison between HTN cases and control.

SNP	Chr (Gene)	Minor Allele	Minor Allele Frequency (%)		*p* Value
			Cases = 200	Control = 200	
rs4536	8 (*CYP11B2*)	T	10	9.0	0.343
rs4537	8 (*CYP11B2*)	C	23	22	0.492
rs4545	8 (*CYP11B2*)	T	8.0	5.0	0.483
rs4543	8 (*CYP11B2*)	T	4.0	3.0	0.432
rs1799998	8 (*CYP11B2*)	T	42	30	0.001
rs4539	8 (*CYP11B2*)	C	34	35	0.879
rs4546	8 (*CYP11B2*)	A	25	23	0.042
rs6418	8 (*CYP11B2*)	G	6.0	5.0	0.462

Reference genome assembly: hg19/GRCh37; Abbreviations: SNP: single nucleotide polymorphism: Chr; chromosome.

**Table 5 genes-14-01184-t005:** Association between selected WES identified SNPs and HTN.

SNP	Chr (Gene)	Minor Allele	OR	CI (95%)	*p* Value
rs4536	8 (*CYP11B2*)	T	0.124	1.15–1.27	0.321
rs4537	8 *(CYP11B2*)	C	0.246	1.54–1.75	0.451
rs4545	8 (*CYP11B2*)	T	0.076	1.12–1.42	0.354
rs4543	8 (*CYP11B2*)	T	0.045	1.53–1.78	0.352
rs1799998	8 (*CYP11B2*)	T	2.257	1.76–2.85	0.045
rs4539	8 (*CYP11B2*)	C	0.754	1.25–1.86	0.765
rs4546	8 (*CYP11B2*)	C	0.053	1.06–1.76	0.071
rs6418	8 (*CYP11B2*)	G	0.872	1.26–2.02	0.327
When adjusted for age, gender, smoking and family history of hypertension					
rs4536	8 (*CYP11B2*)	T	0.214	1.32–1.524	0.348
rs4537	8 (*CYP11B2*)	C	0.112	1.05–1.87	0.254
rs4545	8 (*CYP11B2*)	T	0.056	1.11–1.47	0.483
rs4543	8 (*CYP11B2*)	T	0.035	1.45–1.75	0.434
rs1799998	8 (*CYP11B2*)	T	2.275	1.75–2.96	0.001
rs4539	8 (*CYP11B2*)	C	0.675	1.15–1.75	0.879
rs4546	8 (*CYP11B2*)	C	0.043	1.05–1.78	0.042
rs6418	8 (*CYP11B2*)	G	0.765	1.15–2.01	0.315

Reference genome assembly: hg19/GRCh37; Abbreviations: SNP: single nucleotide polymorphism: Chr; chromosome: OR; odd ratio: CI; confidence interval.

## Data Availability

All necessary information is provided along with the manuscript. However, for further information related to this article, the corresponding author can be contacted.

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
