# Peer review of "Association between Aldosterone Synthase (CYP11B2) Gene Polymorphism and Hypertension in Pashtun Ethnic Population of Khyber Pakhtunkwha, Pakistan"

_genes, 2023, doi:10.3390/genes14061184_

Round 1
Reviewer 1 Report
Dear Authors
The paper is very interesting because it deals with a very important health problem such as hypertension.
I would like to make some comments and raise some doubts.
Due to my academic background, I will only focus on the methodology and data analysis part.
Major Issue
Although I believe that the sample is large enough to have good results, I would like to know if a sample size study was done and how it was done so that this sample is representative of the study population. If this sample size study was not done, I would like to know if a statistical power study was done for a sample of this size.
Minor issue
In the text the term "gender" is used when the variable is "sex". The term "gender" refers to social role and how society views you. For this reason I consider it necessary to use the term "sex" throughout the paper.
The Standard Deviation (SD) is a measure of dispersion, so when it is accompanied by the ± sign it can be confusing. This ± sign is accompanied by the Standard Error (SE) since this error is a measure of precision. I suggest using Mean (SD) or Mean ± SE.
- Altman DG, Gore SM, Gadner MJ, Pocock SJ. Statistical guidelines for contributors to medical journals. Br Med J 1983;286: 1,489-1,493.
- Bailar JC, Mosteller F. Guidelines for statistical reporting in articles for medical journals: amplifications and explanations. Ann Intern Med 1988;108: 266-273.
- Tobias A. [Mean +/- SD, an incorrect expression].Med Clin (Barc). 1998 Feb 7;110(4):157.
In the text they speak that the W Shapiro-Wilk was done when the only continuous variable is age and I think that for this case it would not have been necessary to do this test. The result of this test is not mentioned anywhere in the text.
In all tables and results unify the number of decimal places in the data. For example, the p value should always have 3 decimal places.
Table 1. I think they should put the totals of both cases and controls for the % to make more sense. The frequency or number of cases is also missing.
Table 2. In both cases and controls there is n (f) and it is not clear what it is. In addition, in male 130 and female 70 it is not understood.
Table 3 and 4. Put the totals with respect to how the % is calculated.
Reviewer 2 Report
This paper contains information related to the polymorphism of the CYP11B2 gene associated with hypertension. The findings are interesting, but these findings are not well reflected in the paper.
1. The title is in all capitals. Is it right to the submission rules? And what does "HYPERTENION" mean? Isn't that a typo in "Hypertension"?
2. [Abstract]Please provide full name of "HTN".
3. [Introduction]The first paragraph is too long. should be properly divided. And, this paper is "Article" not "Review". It is necessary to omit excessive details.
4. [Introduction]Gene names should be indicated in italic and should be uniform throughout. (e.g. lines 95, 98, and 116)
5. Please unify Table 1 and Table 2.
6. Figure 1 is not suitable to the level of GENES journal. If possible, it would be good to present the research process as a flowchart. "Synonymous SNVs 30031" -> "Synonymous SNVs 30,031"
7. [Table 5 and Table 6] Which genome version did you use? Although "hg19" is specified in Line 200, it is better to specify the genome version in each table legend.
Reviewer 3 Report
The paper seems somewhat underpowered to test for this kind of effect, as HPTN is a very complex phenotype. Why target only 30 years of age or older? Also, how did you include measurements and medication together? I assumed that you need multiple measurements of increased blood pressure to assess HPTN, how come you managed to do this by only one measurement? How were the 200+200 included, was this a convenience sampling? Also, I am sure that a more refined phenotype measurement could have been performed, including lifestyle and diet estimates, which are critical for this. How did you assess the viral status – self-reported, medication or did you perform the test?
Round 2
Reviewer 2 Report
All points were well revised.
Reviewer 3 Report
Thank you for the comment